# Disparities in Oral Nutritional Supplement Usage and Dispensing Patterns across Primary Care in Ireland: ONSPres Project

**DOI:** 10.3390/nu14020338

**Published:** 2022-01-14

**Authors:** Aisling A. Geraghty, Laura McBean, Sarah Browne, Patricia Dominguez Castro, Ciara M. E. Reynolds, David Hanlon, Gerard Bury, Margaret O’Neill, Sarah Clarke, Barbara Clyne, Karen Finnigan, Laura McCullagh, Sharon Kennelly, Clare A. Corish

**Affiliations:** 1School of Public Health, Physiotherapy and Sports Science, University College Dublin, Belfield, Dublin 4, Ireland; aisling.geraghty@ucdconnect.ie (A.A.G.); laura.mcbean@ucdconnect.ie (L.M.); sarah.browne1@ucd.ie (S.B.); domingup@tcd.ie (P.D.C.); cmereynolds2@gmail.com (C.M.E.R.); 2UCD Institute of Food and Health, University College Dublin, Belfield, Dublin 4, Ireland; 3National HSE Primary Care Division, Service Improvement, Mountmellick Primary Care Building, Co. Laois, Ireland; david.hanlon1@hse.ie (D.H.); sharon.kennelly@hse.ie (S.K.); 4School of Medicine, University College Dublin, Belfield, Dublin 4, Ireland; gerard.bury@ucd.ie; 5National HSE Health and Wellbeing Division, Dr. Steevens’ Hospital, Dublin 8, Ireland; margaret.oneill1@hse.ie; 6HSE Medicines Management Programme, Department of Pharmacology and Therapeutics, Trinity Centre for Health Sciences, St. James’s Hospital, Dublin 8, Ireland; saclarke@stjames.ie; 7HRB Centre for Primary Care Research, Department of General Practice, RSCI University of Medicine and Health Sciences, 123 St. Stephens Green, Dublin 2, Ireland; barbaraclyne@rcsi.ie; 8Department of Pharmacology and Therapeutics, Trinity Centre for Health Sciences, St. James’s Hospital, Dublin 8, Ireland; karen.finnigan@tcd.ie (K.F.); lmccullagh@stjames.ie (L.M.)

**Keywords:** malnutrition, oral nutritional supplements, primary care, older adults

## Abstract

When treating malnutrition, oral nutritional supplements (ONSs) are advised when optimising the diet is insufficient; however, ONS usage and user characteristics have not been previously analysed. A retrospective secondary analysis was performed on dispensed pharmacy claim data for 14,282 anonymised adult patients in primary care in Ireland in 2018. Patient sex, age, residential status, ONS volume (units) and ONS cost (EUR) were analysed. The categories of ‘Moderate’ (<75th centile), ‘High’ (75th–89th centile) and ‘Very High’ ONS users (≥90th centile) were created. The analyses among groups utilised *t*-tests, Mann–Whitney U tests and chi-squared tests. This cohort was 58.2% female, median age was 76 years, with 18.7% in residential care. The most frequently dispensed ONS type was very-high-energy sip feeds (45% of cohort). Younger males were dispensed more ONSs than females (<65 years: median units, 136 vs. 90; *p* < 0.01). Patients living independently were dispensed half the volume of those in residential care (112 vs. 240 units; *p* < 0.01). ‘Moderate’ ONS users were dispensed a yearly median of 84 ONS units (median cost, EUR 153), ‘High’ users were dispensed 420 units (EUR 806) and ‘Very High’ users 892 yearly units (EUR 2402; *p* < 0.01). Further analyses should focus on elucidating the reasons for high ONS usage in residential care patients and younger males.

## 1. Introduction

In Europe, 8.5% of older adults in the community and 17.5% in residential care are at high risk of malnutrition [1]. Approximately 140,000 adults in Ireland have disease-related malnutrition [2]. Enhancing dietary intake is the first approach for treatment of malnutrition and involves advice to consume foods that are high in energy and protein [3]. In circumstances where optimising the diet is not sufficient and malnutrition risk is high, oral nutritional supplements (ONSs) are advised. ONSs are products formulated to provide an energy- and protein-dense addition to an individual’s habitual diet without the suppression of appetite or food intake [4]. ONSs are commercially manufactured and can be in liquid form, semi-solids, or powders, containing varying concentrations of macronutrients and micronutrients. Patients prescribed ONSs should be reviewed within three months to assess their clinical condition and monitor progress [5,6]. However, research in Ireland and the UK indicates that many patients prescribed ONSs do not undergo regular review [7,8,9].

General practitioners (GPs) are frequently the first contact point for community-dwelling individuals at risk of malnutrition and are the primary prescribers of ONSs in Ireland [10]. However, GPs often feel unsupported when managing malnutrition and report a lack of evidence-based decision making and confidence around ONSs [11]. In the UK, inconsistency and substantial differences in both the identification and management of malnutrition amongst adults in the community is also an issue [12]. ONS usage is influenced by an array of factors, such as variation in healthcare professionals’ approaches, poor monitoring and inefficiencies as patients move between healthcare settings [11,13,14]. In the UK’s National Health Service, dietitians can also prescribe ONSs, but prescribing is still influenced by various factors, including local policy and protocol, and patient circumstances [14].

Despite research focusing on determinants of malnutrition and methods of management [15], there remains a dearth of knowledge on patterns of ONS usage in the community and characteristics of ONS users. We aim to investigate the characteristics of patients in receipt of ONSs and describe patterns of ONS dispensing in a large population in Ireland in 2018.

## 2. Materials and Methods

### 2.1. Study Design

A retrospective secondary analysis was performed on anonymised dispensed pharmacy claim data in 2018 (from 1 January to 31 December inclusive). The data were obtained from the General Medical Services scheme (GMS), which provides free at the point-of-delivery general practice care to around 30% of the Irish population with a defined low-income threshold. A database was obtained which contained prescriptions dispensed through the GMS scheme, alongside demographic information for patients and prescribers (GPs). This information is managed by the Health Service Executive Primary Care and Eligibility Reimbursement Service (PCRS). The study examined all dispensed claims originating from GP practices located in three of Ireland’s nine Community Health Organisations (in the counties of Dublin, Kildare and Wicklow in the Republic of Ireland) which comprise 30% of all dispensed pharmacy claims and 33% of the GMS population nationally (approximately 300,000 people).

Data relating to all non-disease-specific ONSs, in addition to anonymised demographic information for 14,282 patients aged ≥18 years and 700 GPs who had prescribed ONSs, were available for 2018. Data were only available for patients or GPs linked to a dispensed claim on the PCRS system. Ethical approval was obtained from the University College Dublin Human Research Ethics Committee (reference LS-18-50-Corish) and the Irish College of General Practitioners (ICGP) Research Ethics Committee.

The following data were analysed: patient sex, age, residential care status, ONS product and volume (in units) dispensed and cost of ONSs (EUR). The data were further grouped by age category (18–44, 45–64 and ≥65 years, in line with previous research in this population [16]) and living situation (patient living in residential care or independently). Non-disease-specific ONSs were categorised by protein and energy content as described in Appendix A. Patients were categorised based on total volume (units) of ONSs dispensed into ‘Moderate’ (<75th centile of volume), ‘High’ (75th–89th centile) and ‘Very High’ ONS users (≥90th centile). This identified high users based on annual volumes dispensed over the course of the year and so may not have identified high clinical or daily usage over a shorter period within the year. Costs refer to the individual product cost calculated using publicly available standardised HSE-listed reimbursement price.

### 2.2. Statistical Analysis

The assessment of normality for all variables was performed by visual analyses of histograms. Parametric or non-parametric tests were used as required. A bivariate analysis was used with one-sample *t*-tests to assess sex differences in the cohort. Mann–Whitney U tests were used to examine sex differences and differences between residential care-dwelling and independent-dwelling patients in relation to age, ONS units and cost. Differences between age groups, intakes of ONS categories between groups and ONS user categories were investigated using cross-tabulations and the chi-squared statistical test. Results with a *p* < 0.05 were considered significant. Statistical analyses were performed using SPSS (Statistical Package for the Social Sciences) software version 24.0 (IBM).

## 3. Results

### 3.1. Characteristics of Patients on Oral Nutritional Supplements

A total of 14,282 community-dwelling adults was dispensed ONSs (Table 1). The median age was 76 years with 71% of the cohort aged over 65 years. Females were older with a median age of 80 years, compared to 71 years in males (*p* < 0.001). In total, 81% of patients dispensed ONSs were living independently and 18.7% were in residential care (*p* < 0.001). A higher proportion of females in residential care were on ONSs than males (23% vs. 12.6%; *p* < 0.001).

There were 1027 GPs on the GMS register within the healthcare areas analysed, 68.2% of whom were linked to a dispensed ONS claim (*n* = 700). Almost one-third (31.8%, *n* = 327) prescribed no ONS during 2018. Per GP, the median number of patients dispensed ONSs was 13; however, this ranged from 1 to 297 and almost 40% of GPs prescribed ONSs to a patient in residential care (Table 1).

### 3.2. Characteristics of Oral Nutritional Supplement Dispensing

A total of 3,640,446 units of ONSs were dispensed to this cohort in 2018. Per patient, this ranged from 1 to 7206 units, with a median of 126 units per patient (Table 2). Most patients (92.5%) were dispensed under 730 units of ONSs, the equivalent of two units per day over the year. Overall, male patients were dispensed higher volumes of ONSs than females; however, when split by age, this was only seen in males aged <65 years (median, 136 units for males vs. 90 units for females; *p* < 0.001). Specifically, males in the 18–44-year-old category were dispensed more units of ONSs than females in the same age category (median units, 120 (IQR 236) for males and 60 (IQR 146) for females; *p* < 0.01).

There were seven ONS categories, with the most common being very-high-energy sip feeds (which were dispensed to 45% of the cohort and making up 31.8% of all products (Table 2, Figure 1)). There was no difference in category of product dispensed to males and females, apart from high-energy semi-solid ONS products, which were more common for females (18% vs. 13.8%; *p* < 0.001). However, when split by age, differences were observed (Table 2; *p* < 0.001).

### 3.3. Oral Nutritional Supplement Dispensing in Residential Care Compared to Independent Living

More patients within residential care were female (72% vs. 55% males) and aged ≥65 years (99% of females vs. 65% of males; *p* < 0.001) than patients living independently (Table 3). Residential care patients had higher volumes of ONS dispensed (median, 240 units vs. 112; *p* < 0.001) and ONSs were also more costly, with the median yearly cost being EUR 541, compared to EUR 212 for patients living independently (*p* < 0.001). Dispensing patterns of all ONS categories differed based on residential status (Table 3). The ONS products dispensed in each category split by residential status are presented in Figure 2.

### 3.4. Characteristics of ‘High’ and ‘Very High’ Oral Nutritional Supplement Users

There were 2152 patients (15.1%) classified as ‘High’ ONS users and 1428 (10%) ‘Very high’ ONS users (Table 4). The median age of those in the ‘High’ and ‘Very High’ user categories was higher than that of ‘Moderate’ ONS users (*p* < 0.001). There were more females in each group; however, the proportion of males increased in the ‘High’ and ‘Very High’ users groups (*p* = 0.04).

Of ‘Very High’ and ‘High’ users, 36.4% and 26.6% were in residential care, respectively, compared to only 14.8% of ‘Moderate’ users (*p* < 0.001). ‘Moderate’ ONS users had a median 84 units yearly of ONS dispensed (median costs of EUR 153 per patient), ‘High’ ONS users 420 units (EUR 806) and ‘Very High’ ONS users had 892 units (EUR 2402; *p* < 0.001).

## 4. Discussion

### 4.1. Main Findings

In this analysis of 14,282 patients aged over 18 years, clear disparities were identified in ONS usage across patient age groups, between males and females and also between patients living independently in the community and patients in residential care facilities. In particular, higher ONS usage was noted in younger males and among patients in residential care, with further differences in ONS category usage identified among patients in residential care. In this sample of one third of Ireland’s population, two-thirds of GPs had prescribed ONSs at least once during the year, with medians of 20 prescriptions for 13.5 patients, identifying this role as an important issue for general practice.

### 4.2. Comparison with Existing Literature

We identified a higher proportion of female patients on ONSs, similar to previous research in Ireland and internationally reporting ONS users being predominantly female and older [1,8]. This may be due to females having higher life expectancies and, as a result, increased frailty and risk of malnutrition. Although more ONS users were female, younger males were dispensed much higher volumes. We found males aged under 44 were dispensed twice the volume of ONSs compared to females, beyond what could be explained by their increased nutritional requirements or treatment of conditions such as Crohn’s Disease. Research carried out in America similarly found that the proportion of males was higher among ONS users in a slightly younger population [17].

This raises the question around the use of ONSs in younger populations and factors influencing their prescription. Our group has previously described an association between ONS prescribing in this age group and simultaneous prescribing of psychoactive drugs [16]; chronic illness and multimorbidity are common among opiate-dependent patients in Ireland [18]. ONS prescribing may be a feature of the management of opiate dependency in Ireland and further research should explore the implications of this finding. Recent qualitative research exploring factors influencing ONS prescribing focused mainly on older adults and found that ONSs were often prescribed without evidence for their use [11,13]. Additional enquiry is needed to understand these patterns and whether they are related to social factors.

Despite the reported effectiveness of ONSs in managing malnutrition in patients in residential care [19], we found that these patients were dispensed twice the volume of ONSs compared to patients living independently. Research carried out on 23,500 patients in residential care across Europe and America found that almost 14% used ONSs and usage was associated with increasing age and functional impairment [20]. Similarly, in this population, polypharmacy was associated with long-term usage of ONSs, which may indicate a decline in health [16]. Although higher requirements for ONSs in residential care may be explained by the advancing age of patients, which is also reflected by the higher proportions of female patients who have a longer life expectancy and higher disease rates, given that they have access to and assistance with meals and snacks in residential care, this increased need should be mitigated. Unfortunately, data on medication or health status were not available in this analysis, but future analyses should incorporate these to help elucidate the reasons behind increased ONS use in residential care.

While dietary counselling combined with ONSs in care homes have been shown to be highly effective for treating malnutrition [21], the appropriateness of ONS prescribing in this setting remains a concern. Reported ONS usage in residential care varies drastically with a recent publication estimating rates ranging from 1% to 43% across countries [20]. Previous research in Ireland indicated that up to one-third of ONS prescriptions in the community were inappropriate for patient needs [8] and a recent review of ONS usage within care homes in the UK identified high levels of inappropriate prescribing [7]. Similarly, a recent analysis in the United States reported that, although 25% of their cohort of adults in the community were classified as being at high risk of malnutrition, only 11% consumed ONSs [17]. This highlights the need for clarification around appropriate prescribing of ONSs, particularly for patients in residential care due to the variations in usage in this setting.

There is limited research on factors influencing prescriber choice of ONS products. However, recent research in Ireland similarly identified a preference for high-protein ONS products with the main factors driving ONS product choice being nutritional value and patient palatability [22]. We found that the ONS category did not differ between males and females; however, patient age and residential location did differ with the ONS category dispensed. Previously, in Ireland, the cost of ONSs was reported as a concern for ONS prescribers [11], which may influence the product prescribed. We found that almost one-third of GPs within these community areas did not prescribe ONSs for any patients, potentially also due to cost concerns and lack of standardised screening for malnutrition. This is despite evidence from a recent systematic review indicating that ONS use in the community is near neutral or produces a cost advantage [23]. This, alongside differences in ONS products and volumes, suggests that education and support are needed for GPs and healthcare professionals prescribing ONSs.

### 4.3. Strengths and Limitations

This analysis had many strengths, in that it included comprehensive prescribing data for around 300,000 people (30% of the GMS population in Ireland) and identified almost 15,000 ONS users. As the data related to dispensed ONS claims, this removed any reporting or recall bias from patients or healthcare providers relating to ONS usage. However, this analysis was not without limitations. Only data relating to patients or GPs linked to a dispensed ONS claim on the PCRS system were available, so comparisons between non-ONS using patients or non-ONS prescribing GPs were not possible. Limited characteristics and demographic information were available on patients and the background of the users, including the purpose of dispensing, is not known, so future analysis should explore the impact of factors such as socioeconomic status, health status and other medication usage. Additional avenues of exploration which could help elucidate these findings include prevalent pathologies, the degree of malnutrition and ONS effectiveness. As these data related to ONS dispensing claims rather than consumption of ONSs, it was not possible to access appropriate ONS usage and, similarly, prescribing practises could not be evaluated without a review of individual patients, which is highly encouraged in future studies.

### 4.4. Implications for Research and Practice

Particular consideration should be given to younger males and patients in residential care facilities which used higher volumes of ONSs. Of equal concern is the possible ‘under-treatment’ of malnutrition by GPs. Given that 30% of GPs did not prescribe any ONSs, there is a need for consistent approaches whereby evidence-based prescribing is implemented in all settings. Further education around appropriate usage of ONSs, both for patients and healthcare professionals, is vital to ensure both appropriate prescribing and appropriate usage of ONSs, to efficiently prevent and treat malnutrition in the community. Multidisciplinary teams are indispensable in effectively managing malnutrition in our community and adequate support must be provided at all levels of healthcare to ensure successful treatments.

## 5. Conclusions

Strong disparities were identified in ONS usage between patient groups in primary care in Ireland. Further research is warranted to elucidate the reasons for high ONS usage in younger males and patients in residential care. Further education and support are required for healthcare professionals working in malnutrition management.

## Figures and Tables

**Figure 1 nutrients-14-00338-f001:**
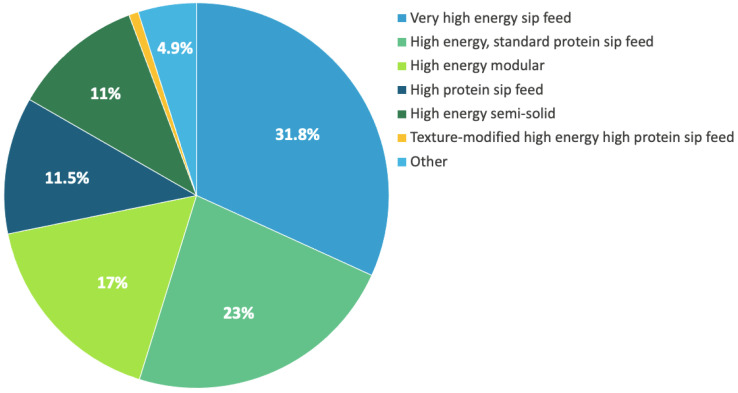
Breakdown of oral nutritional supplement products dispensed across categories in the full cohort (*n* = 14,282).

**Figure 2 nutrients-14-00338-f002:**
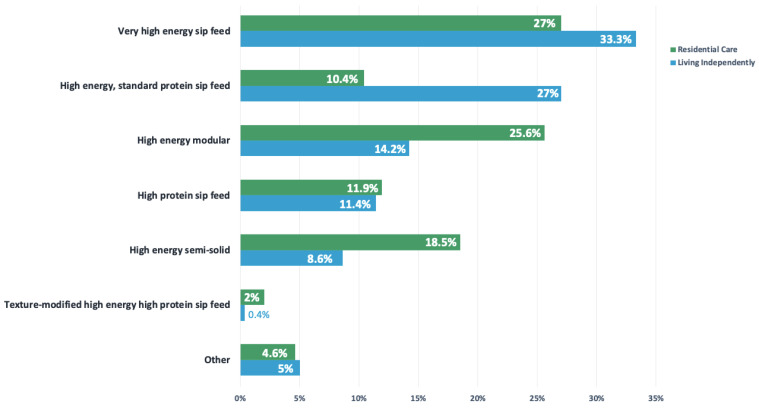
Bar chart with breakdown of all oral nutritional supplement products dispensed in the cohort from each category to patients in residential care and patients living independently.

**Table 1 nutrients-14-00338-t001:** Characteristics of patients dispensed oral nutritional supplements and their general practitioners over a 12-month period.

	Total		Male		Female		*p*
Patients (*n*, %)	14,282	100	5946	41.6	8336	58.5	<0.001 ^a^
Age (years, median, IQR)	76.0	24.0	71.0	27.0	80.0	20.0	<0.001 ^b^
18–44 years (*n*, %)	1500	10.5	784	13.2	716	8.6	<0.001 ^c^
45–64 years (*n*, %)	2592	18.2	1449	24.4	1143	13.7	
≥65 years (*n*, %)	10,190	71.4	3713	62.5	6477	77.7	
**Patient Care Location**							
Residential care (*n*, %)	2674	18.7	750	12.6	1934	23.1	<0.001 ^c^
Independent living (*n*, %)	11,608	81.3	5196	87.4	6412	76.9	
**General Practitioners (*n* = 700)**							
Number of patients on ONSs (median, IQR)				13.5	22.0		
Treating patients in residential care (*n*, %)				273	39.0		
Total ONS prescriptions † per GP (median, range)				20	1–683		
Prescriptions † per patient (median, range)				1.43	1–12		

IQR, interquartile range; ONS, oral nutritional supplement. Non-parametric data are reported as median, IQR. Statistical tests: ^a^ One sample *t*-test, ^b^ Mann–Whitney U test, ^c^ chi-squared test. Statistical significance set at *p* < 0.05. Data refer to non-disease-specific oral nutritional supplements dispensed between 1 January and 31 December 2018 inclusive. † Prescription related to one ONS product and irrespective of volume.

**Table 2 nutrients-14-00338-t002:** Characteristics of oral nutritional supplement usage over a 12-month period.

	Total		Males		Females		*p*
**ONS Volume (Units)**							
Units of ONSs per patient (entire cohort) (median, IQR)	126.0	262.0	140.0	274.0	120.0	256.8	0.005 ^b^
Age group: 18–64 years (median, IQR)	112.0	244.0	136.0	293.0	90.0	210.0	<0.001 ^b^
Age group: ≥65 years (median, IQR)	140.0	268.0	140.0	252.0	140.0	278.5	0.739 ^b^
**ONS Costs (EUR)**							
Cost of ONS per patient (entire cohort) (median, IQR)	251	585	265	604	246	580	0.007 ^b^
Age group: 18–64 years (median, IQR)	214	524	255	610	179	416	<0.001 ^b^
Age group: ≥65 years (median, IQR)	269	627	269	596	269	654	0.937 ^b^
**ONS Category**	**Total Cohort** ***n* = 14,282**		**Patients** **aged 18–64 years** ***n* = 4092**		**Patients** **aged ≥ 65 years** ***n* = 10,190**		** *p* **
Very-high-energy sip feed (*n*, %)	6459	45.2	1773	43.3	4686	46.0	0.004 ^c^
High-energy, standard protein sip feed (*n*, %)	4829	33.8	1813	44.3	3016	29.6	<0.001 ^c^
High-energy modular (*n*, %)	3393	23.8	617	15.1	2776	27.2	<0.001 ^c^
High-protein sip feed (*n*, %)	2473	17.3	611	14.9	1862	18.3	<0.001 ^c^
High-energy semi-solid (*n*, %)	2324	16.3	382	9.3	1942	19.1	<0.001 ^c^
Texture-modified high-energy, high-protein sip feed (*n*, %)	168	1.2	10	0.2	158	1.6	<0.001 ^c^
Other (*n*, %)	1056	7.4	288	7.0	769	7.5	0.303 ^c^

IQR, interquartile range; ONS, oral nutritional supplement. Non-parametric data are reported as median, IQR. Statistical tests: ^b^ Mann–Whitney U test, ^c^ chi-squared test. Significance set at *p* < 0.05. Data refer to non-disease-specific oral nutritional supplements dispensed between 1 January and 31 December 2018 inclusive.

**Table 3 nutrients-14-00338-t003:** Comparison of characteristics and oral nutritional supplement dispensing between patients living in residential care and living independently across a 12-month period.

	Residential Care		Independent-Living		*p*
Total (*n*, %)	2674	18.7	11,608	81.3	
Female (*n*, %)	1924	72.0	6412	55.2	<0.001 ^a^
Age (years, median, IQR)	86.0	10.0	73.0	25.0	<0.001 ^b^
18–44 years (*n*, %)	4	0.1	1496	12.9	<0.001 ^a^
45–64 years (*n*, %)	15	0.6	2577	22.2	
≥65 years (*n*, %)	2655	99.3	7535	64.9	
**ONS**					
Units per patient (median, IQR)	240	436	112	224	<0.001 ^b^
Cost per patient (EUR, median, IQR)	541	1398	212	479	<0.001 ^b^
**ONS Category**					
Very-high-energy sip feed (*n*, %)	1290	48.2	5169	44.5	0.001 ^a^
High-energy, standard protein sip feed (*n*, %)	513	19.2	4316	37.2	<0.001 ^a^
High-energy modular (*n*, %)	1194	44.7	2199	18.9	<0.001 ^a^
High-protein sip feed (*n*, %)	603	22.6	1870	16.1	<0.001 ^a^
High-energy semi-solid (*n*, %)	938	35.1	1386	11.9	<0.001 ^a^
Texture-modified high-energy, high-protein sip feed (*n*, %)	105	3.9	63	0.5	<0.001 ^a^
Other (*n*, %)	235	8.8	821	7.1	0.002 ^a^

IQR, interquartile range; ONS, oral nutritional supplement. Non-parametric data are reported as median, IQR. Statistical tests: ^a^ chi-squared test, ^b^ Mann–Whitney U test. Significance set at *p* < 0.05. Data refer to non-disease-specific oral nutritional supplements dispensed between 1 January and 31 December 2018 inclusive.

**Table 4 nutrients-14-00338-t004:** Characteristics of ‘Moderate’, ‘High’ and ‘Very High’ volume oral nutritional supplement usage in cohort over a 12-month period.

	‘Moderate’ ONS Users (<75th Centile)		‘High’ ONS Users (75th–89th Centile)		‘Very High’ ONS Users (≥90th Centile)		*p*
Patients (*n*, %)	10,702	74.9	2152	15.1	1428	10	
Male (*n*, %)	4425	41.3	909	42.2	612	42.9	0.040 ^a^
Female (*n*, %)	6277	58.7	1243	57.8	816	57.1	
Age (years, median, IQR)	76	24	78	24	77	25	<0.001 ^b^
18–44 years (*n*, %)	1160	10.8	197	9.2	143	10.0	0.022 ^a^
45–64 years (*n*, %)	1942	18.1	378	17.6	272	19.0	
≥65 years (*n*, %)	7600	71.0	1577	73.3	1013	70.9	
**Patient Care Location**							
Residential care (*n*, %)	1581	14.8	573	26.6	520	36.4	0.001 ^a^
Independent living (*n*, %)	9121	85.2	1579	73.4	908	63.6	
**ONS**							
Units per patient (median, IQR)	84	127	420	146	892	476	<0.001 ^b^
Cost per patient (EUR, median, IQR)	153	240	806	490	2402	6347	<0.001 ^b^

IQR, Interquartile range; ONS, oral nutritional supplement. Non-parametric data are reported as median, IQR. Statistical tests: ^a^ chi-squared test, ^b^ Mann–Whitney U test Significance set at *p* < 0.05. Data refer to non-disease-specific oral nutritional supplements dispensed between 1 January and 31 December 2018 inclusive.

## Data Availability

Data available upon reasonable request to the corresponding author.

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
