# Peer review of "Disparities in Oral Nutritional Supplement Usage and Dispensing Patterns across Primary Care in Ireland: ONSPres Project"

_nutrients, 2022, doi:10.3390/nu14020338_

Round 1
Reviewer 1 Report
This article is a very simple descriptive statistical study based on dispensing data. It is a very interesting point of view, but there are many limitations to the study, such as the limited scope of the study and the fact that the true background of the users, such as the purpose of dispensing, is not known. Nevertheless, as the author wrote, I believe that this will provide valuable basic information for research to be conducted in the future.
Author Response
Comments and Suggestions for Authors
This article is a very simple descriptive statistical study based on dispensing data. It is a very interesting point of view, but there are many limitations to the study, such as the limited scope of the study and the fact that the true background of the users, such as the purpose of dispensing, is not known. Nevertheless, as the author wrote, I believe that this will provide valuable basic information for research to be conducted in the future.
Response: Many thanks for the helpful and supportive feedback. We agree with your point about the limitations of this work and to ensure this is clear we have updated the limitations that the backgrounds of the users are unknown which would require further analyses in the future: “Limited characteristics and demographic information were available on patients and the background of the users, including as the purpose of dispensing, is not known so future analysis should explore the impact of factors such as socioeconomic status, health status, and other medication usage. Additional avenues of exploration which could help elucidate these findings include prevalent pathologies, the degree of malnutrition and, ONS effectiveness.” (line 514-519)
Reviewer 2 Report
This represents an interesting survey of ONS prescription in Ireland. But it provides a rather limited insight given the restricted detail available for consideration. This shortcoming is rightly acknowledged in the Discussion
Clear and well composed, and appears well informed, and is well discussed.
Nicely compares findings with related data in other nations, with similar trends noted.
Issues of concern are raised that should assist introduction of corrective actions to improve ONS prescription practices.
Specific comments
Title - Does ‘Ireland’ refer to the Republic and Northern Ireland? This will not be clear to many outside Ireland I think
Intro – it would be useful to provide a little more info on the exact composition of ONS, and to indicate whether there are differences between ONS sources – as supplied by different manufacturers etc or recommended for different purposes (e.g. high/low energy) - I note some info on this appears in the Results.
Table 2 – which of the 2 values do the P values refer to – median or IQR? This issue applies to other Tables also
162-4 - does the higher level of females taking ONS in residential care reflect the higher numbers of females in residential care? For this value to be of any use then it is necessary to know the proportion of males/females in residential care taking ONS
30 – 135 or 136? – see table 2!
85-86 – why have these particular age groups been selected? Please provide a justification
130-2 – these data are missing from Table 2 – please include
Fig 2 - lacks title and legend
216 – here, please point to the relevant data
227 – but you provide no specific info on males under the age of 40. The relevant age range utilised 18-44; please be consistent
Author Response
Comments and Suggestions for Authors
This represents an interesting survey of ONS prescription in Ireland. But it provides a rather limited insight given the restricted detail available for consideration. This shortcoming is rightly acknowledged in the Discussion
Clear and well composed, and appears well informed, and is well discussed.
Nicely compares findings with related data in other nations, with similar trends noted.
Issues of concern are raised that should assist introduction of corrective actions to improve ONS prescription practices.
Specific comments
Title - Does ‘Ireland’ refer to the Republic and Northern Ireland? This will not be clear to many outside Ireland I think
Response: Many thanks for your helpful comments. We included Ireland in the title to give the readers a quick insight into the population being studied, however, this is a subset analysis of patients across 3 Counties in the Republic of Ireland. We have made this clearer in the methods (line 238-242)
“The study examined all dispensed claims originating from GP practices located in three of Ireland’s nine Community Health Organisations (in Counties Dublin, Kildare, and Wicklow in the Republic of Ireland) which comprise 30% of all dispensed pharmacy claims and 33% of the GMS population nationally (approximately 300,000 people).”
Intro – it would be useful to provide a little more info on the exact composition of ONS, and to indicate whether there are differences between ONS sources – as supplied by different manufacturers etc or recommended for different purposes (e.g. high/low energy) - I note some info on this appears in the Results.
Response: Many thanks for your comment, we have updated the introduction to provide more background on ONS products: “ONS are products formulated to provide an energy- and protein-dense addition to an individual's habitual diet without the suppression of appetite or food intake [4]. ONS are commercially manufactured and can be in liquid form, semi-solids, or powders, containing varying concentrations of macronutrients and micronutrients.“ (see line 45-209).
Table 2 – which of the 2 values do the P values refer to – median or IQR? This issue applies to other Tables also
Response: Thank you for this comment. The P values differ depending on the corresponding tests that were carried out, in this case the Mann-Whitney U test. For these data the median and IQR are given to describe the distribution of the data to aid interpretation for the readers. The Mann-Whitney U test is about the distributions and uses the ranks. It does not compare median or IQR. We noted in each table which test was carried out for each P value for clarity. We have checked and edited each table to ensure that this information is clear for each P value (See tables 2, 3, and 4)
162-4 - does the higher level of females taking ONS in residential care reflect the higher numbers of females in residential care? For this value to be of any use then it is necessary to know the proportion of males/females in residential care taking ONS
Response: Thank you for this comment. At the start of the results section and in table 1 we describe the proportion of females in residential care that were on ONS compared with males (23% vs 12.6%, P<0.001). There generally are more female patients in residential care due to longer life expectancies and we have updated our discussion to reflect this: “Although higher requirements for ONS in residential care may be explained by advancing age of patients, which is also reflected by the higher proportions of female patients who have a longer life expectancy, and higher disease rates, given that they have access to and assistance with meals and snacks in residential care, this increased need should be mitigated.” (line 464-468).
30 – 135 or 136? – see table 2!
Response: Thank you for this comment. Apologies for the typo, this has been fixed (See abstract, line 30)
85-86 – why have these particular age groups been selected? Please provide a justification
Response: Thank you for this comment. These age categories were selected in keeping with previous research in this population. We have added this information into the methods and included the reference (see line 251).
130-2 – these data are missing from Table 2 – please include
Response: Thanks for this comment. We included the information “Most patients (92.5%) were dispensed under 730 units of ONS, the equivalent of two units per day over the year” just to give some clinical context to the ONS units. This was not analysed any further and so authors decided not to include in Table 2 for clarity, however this can be added if needed.
Fig 2 - lacks title and legend
Response: Thank you for this comment. Apologies for this omission, the title has now been added: Figure 2: Bar chart with breakdown of all oral nutritional supplement products dispensed in the cohort from each category to patients in residential care and patients living independently (see lines 393-394)
216 – here, please point to the relevant data
Response: Thank you for this comment. Here we had referred to differences in ONS category “preference” identified among patients in residential care, however, we should have stated differences in ONS category “usage”. We have rectified this (see line 426).
227 – but you provide no specific info on males under the age of 40. The relevant age range utilised 18-44; please be consistent
Response: Thank you for this comment and we apologise for the typo here. We have changed this in the text to males under the age of 44 – many thanks for highlighting this. (See line 443).
Reviewer 3 Report
The work investigate characteristics of ONS users and patterns of ONS dispensing in a large population in Ireland in receipt of ONS in 2018. For this, the following data were collected: patient sex, age, residential status, ONS product and volume (units) dispensed, and cost of ONS (€). The number of patients collected is very important (14,282 patients) but the data collected is not clinically relevant (age, sex and residential status). It would be more interesting to know the pathologies that cause the indication of ONS, the type of malnutrition suffered by patients, the duration of treatment, morbidity and mortality results, etc ...
It is a retrospective secondary analysis on anonymised dispensed pharmacy claims data on the General Medical Services Scheme in Ireland in 2018. The first question that arises to me is that they are indirect data. We do not know the degree of compliance with nutritional treatment.
In summary, the authors collect administrative data, but they are not clinical data. Results may be biased by data collection. It is not expected that community patients present more malnutrition than elderly residents (81% of patients dispensed ONS were living independently and 18.7% were in residential care). On the other hand, they observe a higher proportion of females in residential care were on ONS compared with males (23% vs 12.6%). This is frequent because survival is higher in women and they reach a situation of serious functional deterioration that requires institutionalization more frequently.
The prescription of ONS by GPs is highly variable, as evidenced by the wide range of 1 to 683 ONS prescriptions per GP.
The authors in lines 253-264 speculate on the misuse of ONS in geriatric residences based on different bibliography but not related to their own results.
The authors indicate that future analysis should explore the impact of factors such as socioeconomic status, health status, and other medication usage. I disagree. I understand that the important factors would be the most prevalent pathologies, the degree of malnutrition and, above all, its effectiveness.
The authors pay close attention to the fact that younger males and patients in residential care facilities which used higher volumes of ONS. It is a finding that cannot be explained with the data collected.
Its conclusions are based on future recommendations because its results are very poor.
Author Response
Comments and Suggestions for Authors
The work investigate characteristics of ONS users and patterns of ONS dispensing in a large population in Ireland in receipt of ONS in 2018. For this, the following data were collected: patient sex, age, residential status, ONS product and volume (units) dispensed, and cost of ONS (€). The number of patients collected is very important (14,282 patients) but the data collected is not clinically relevant (age, sex and residential status). It would be more interesting to know the pathologies that cause the indication of ONS, the type of malnutrition suffered by patients, the duration of treatment, morbidity and mortality results, etc ... It is a retrospective secondary analysis on anonymised dispensed pharmacy claims data on the General Medical Services Scheme in Ireland in 2018. The first question that arises to me is that they are indirect data. We do not know the degree of compliance with nutritional treatment.
Response: Many thanks for your helpful feedback. We agree that this is a very important area which warrants further research. Unfortunately, we did not have access to information relating to indication of ONS, patient morbidity, or compliance which we have included in the limitations of the paper: “Limited characteristics and demographic information were available on patients and the background of the users, including as the purpose of dispensing, is not known so future analysis should explore the impact of factors such as socioeconomic status, health status, and other medication usage. Additional avenues of exploration which could help elucidate these findings include prevalent pathologies, the degree of malnutrition and, ONS effectiveness.” (see lines 514-519).
In summary, the authors collect administrative data, but they are not clinical data. Results may be biased by data collection. It is not expected that community patients present more malnutrition than elderly residents (81% of patients dispensed ONS were living independently and 18.7% were in residential care). On the other hand, they observe a higher proportion of females in residential care were on ONS compared with males (23% vs 12.6%). This is frequent because survival is higher in women and they reach a situation of serious functional deterioration that requires institutionalization more frequently.
Response: Many thanks for your helpful comments. We have updated our discussion to reflect this important point: “Although higher requirements for ONS in residential care may be explained by advancing age of patients, which is also reflected by the higher proportions of female patients who have a longer life expectancy, and higher disease rates, given that they have access to and assistance with meals and snacks in residential care, this increased need should be mitigated.” (see lines 464-468).
The prescription of ONS by GPs is highly variable, as evidenced by the wide range of 1 to 683 ONS prescriptions per GP.
Response: Thank you for this comment. We agree that this was an important findings in our analysis which informed our recommendation that further education around appropriate usage of ONS both for patients and healthcare professionals is vital to ensure both appropriate prescribing and appropriate usage of ONS to effectively prevent and treat malnutrition in the community (see lines 529-531).
The authors in lines 253-264 speculate on the misuse of ONS in geriatric residences based on different bibliography but not related to their own results.
Response: Thank you for this comment. We included in our discussion references to findings from other published work which could potentially explain our results in this population in order to put our work into context. Unfortunately, we did not have data available in our cohort to allow us to investigate the reasons behind our findings and so we include in our discussion that “Additional enquiry is needed to understand these patterns and whether they are related to social factors.” (see lines 456-457)
The authors indicate that future analysis should explore the impact of factors such as socioeconomic status, health status, and other medication usage. I disagree. I understand that the important factors would be the most prevalent pathologies, the degree of malnutrition and, above all, its effectiveness.
Response: Thank you for this comment, we agree that these also represent important avenues of investigation and so have included these in our discussion: “Additional avenues of exploration which could help elucidate these findings include prevalent pathologies, the degree of malnutrition and, ONS effectiveness.“ (see lines 518-519).
The authors pay close attention to the fact that younger males and patients in residential care facilities which used higher volumes of ONS. It is a finding that cannot be explained with the data collected.
Its conclusions are based on future recommendations because its results are very poor.
Response: Thank you for this comment. Our analyses found that these demographics were associated with higher usage of ONS although, unfortunately, we lacked additional data and were unable to further explore potential reasons for this in our population. We included this in our limitations and hope that these results will help guide future research in this area with a focus on these groups to elucidate the reasons behind this.
Round 2
Reviewer 3 Report
It is a very well written study but the data on which it is based are of no clinical interest